# RECURSIVE REGRESSION WITH NEURAL NETWORKS: APPROXIMATING THE HJI PDE SOLUTION

**Vicenç Rúbies Royo, Claire Tomlin**
Department of Electrical Engineering and Computer Sciences
UC Berkeley
Berkeley, California, USA
`vrubies@berkeley.edu, tomlin@berkeley.edu`

## ABSTRACT

Most machine learning applications using neural networks seek to approximate some function $g(x)$ by minimizing some cost criterion. In the simplest case, if one has access to pairs of the form $(x, y)$ where $y = g(x)$, the problem can be framed as a regression problem. Beyond this family of problems, we find many cases where the unavailability of data pairs makes this approach unfeasible. However, similar to what we find in the reinforcement learning literature, if we have some known properties of the function we are seeking to approximate, there is still hope to frame the problem as a regression problem. In this context, we present an algorithm that approximates the solution to a partial differential equation known as the Hamilton-Jacobi-Isaacs partial differential equation (HJI PDE) and compare it to current state of the art tools. This PDE, which is found in the fields of control theory and robotics, is of particular importance in safety critical systems where guarantees of performance are a must.

## 1 INTRODUCTION

Artificial neural networks are remarkable function approximators used in a myriad of applications ranging from complex controllers for robotic actuation (Levine et al., 2016) (Schulman et al., 2015) to simple image classifiers for digit recognition (LeCun et al., 1989) . They even find uses in physics to find approximations to solutions of PDEs and systems of coupled ordinary differential equations (ODEs) (Lagaris et al., 1998). Their success is in part achieved by their property of being universal function approximators (Hornik et al., 1989). In order to train a neural network one usually defines a cost function which captures the "goodness" of the choice of parameters in our model, and uses gradient descent/ascent algorithms to improve them. In supervised learning, for example, input output data pairs are used to define a cost function such as the mean squared error or the mean absolute error; unfortunately, in many cases the function we want to approximate is unkown. For instance, in many reinforcement learning settings one wants to find the optimal policy, a function from state variables to actions[1], which maximizes the expected sum of discounted rewards of an agent in some environment. This function is usually unkown *a priori*, so this problem can't readily be framed as a regression problem using input-output pairs. This assertion becomes blurred, however, when looking at the work of Mnih et al. (2013), where a deep Q-network learns by generating targets and minimizing a cost of the form

$$L_i(\theta_i) = \mathbb{E}_{s,a\sim\rho}[(y_i - Q(s, a; \theta_i))^2].\tag{1}$$

Here, the targets $y_i$ are generated from the same Q-network that is being used to approximate the Q-function, hence the neural network has two purposes: approximation and data generation. In this work, we show that this same idea can be extended to the domain of approximating solutions to partial differential equations, and in particular the solution to the Hamiton-Jacobi-Isaacs PDE.

---

[1] or states to probabilities over actions

## 2 THE HAMILTON-JACOBI-ISAACS PDE

In control theory and robotics we often want to know how a system evolves in time given some input signal. In particular, one would like to know whether there exists an (optimal) input signal that drives our system to a particular region of interest in our state space and what that input is. For a deterministic system with continuous states and inputs, this problem can be succinctly expressed as a partial differential equation known as the Hamilton-Jacobi-Isaacs (HJI) PDE.

Let $V : \mathbb{R}^n \times \mathbb{R}^- \to \mathbb{R}$. Then, given a time invariant system of the form $\frac{dx}{dt} = f(x, a, b)$ and boundary condition $V(x, 0) = l(x)$, where $x \in \mathbb{R}^n$ is the state vector and $a \in \mathcal{A} \subseteq \mathbb{R}^{m_a}$ and $b \in \mathcal{B} \subseteq \mathbb{R}^{m_b}$ are inputs to the system[2], we wish to find the solution to the minimum-payoff HJI PDE, associated to the reachability problem:

$$\frac{\partial V(x, t)}{\partial t} = -min\{0, H(x, \nabla_x V)\}, \tag{2}$$

where

$$H(x, \nabla_x V) := \max_{a \in \mathcal{A}} \min_{b \in \mathcal{B}} \nabla_x V^T f(x, a, b) \tag{3}$$

is known as the Hamiltonian. The boundary condition $V(x, 0) = l(x)$ encodes in its zero sub-level set (i.e. $l(x) \leq 0$) the region of interest in our state space known as the target set $\mathcal{T}$. Lastly, the solution $V(x, t)$ to (2) encodes the information about all the starting states whose induced trajectories will enter (and possibly leave) $\mathcal{T}$ within $|t|$, given the dynamics and input signals. More precisely, for some starting state $x_0$ and $t \leq 0$, $V(x_0, t) < 0$ if and only if the trajectory starting from $x_0$ enters $\mathcal{T}$ within $|t|$.

To give some intuition as to why $V(x, t)$ encodes the starting states whose trajectories enter $\mathcal{T}$ within $t$, let us consider the simpler problem where $\frac{dx}{dt} = f(x)$ is an autonomous system without any inputs. Further, let us write (2) as a finite difference in $t$. With some rearranging, and absorbing the gradient into $V$ (i.e. $\nabla_x V^T f(x)\Delta t + V(x, t) \approx V(x + f(x)\Delta t, t)$), one can obtain the following approximation

$$V(x, t - \Delta t) \approx min\{ V(x, t) , V(x + f(x)\Delta t, t) \}. \tag{4}$$

It is straightforward to see from (4) that at time $t = 0$ all the states outside of $\mathcal{T}$ (i.e. $V(x, 0) \geq 0$) but near its boundary, whose induced trajectories enter the target (i.e. $V(x + f(x)\Delta t, 0) < 0$) within $\Delta t$, will become negative in $V(x, -\Delta t)$. Thinking of this update recursively one can intuitively see how the zero sub-level set of $V$ grows backward in time to include more and more states.

For the case of one input trying to drive our system into $\mathcal{T}$, the approximation becomes

$$V(x, t - \Delta t) \approx min\{ V(x, t) , \min_b V(x + f(x, b)\Delta t, t) \}, \tag{5}$$

and for two competing inputs,

$$V(x, t - \Delta t) \approx min\{ V(x, t) , \max_a \min_b V(x + f(x, a, b)\Delta t, t) \}. \tag{6}$$

Using the previous analogy of the autonomous system, one can see how (5) and (6) are essentially different ways to expand the zero sub-level set backward in time: (5) can be seen as an input trying to expand the set as fast as possible; (6) can be seen as two inputs with competing goals, where one input tries to expand the set and the other seeks to prevent its growth. Moreover, this last setting shows the relevance of the HJI PDE in safety critical systems. By treating input $b$ as a bounded worse case disturbance and $\mathcal{T}$ as some unsafe region, one can establish safety guarantees about the system and claim which states won't be driven into $\mathcal{T}$ within some time horizon.

---

[2]$a$ is usually taken to be the input and $b$ is taken to be some bounded input disturbance

Lastly, it is important to note that $V(x, t)$ contains useful information in its gradient $\nabla_x V(x, t)$. In the case where $\frac{dx}{dt} = f(x, b)$ has a single input, the argument minimizing the following optimization problem

$$b^* = \underset{b \in \mathcal{B}}{argmin} \ \nabla_x V(x_o, t)^T f(x_o, b) \tag{7}$$

yields the instantaneous optimal input for state $x_0$ at time $t$ to guide the trajectory into $\mathcal{T}$ as fast as possible. Using this fact one can generate an optimal control policy based on the gradient of $V$. This idea can then be easily extended to the case of two competing inputs to obtain competing control policies. Finally, even though (7) need not be a convex problem, in this work we will only deal with simple dynamical systems, making the optimization problem easy to solve.

## 3  APPROXIMATING SOLUTIONS OF PDEs

The problem presented in section 2 (as in many other cases with PDEs) is general not straightforward to solve. For this reason, trying to find a good approximation instead of the actual solution can be a reasonable approach. Many current state-of-the-art tools used to approximate solutions of PDEs, including (2), use gridding techniques (Mitchell, 2007) whereby finite differences are used to iteratively update values on a grid. Another approach (Lagaris et al., 1998) is to train a feedforward neural network by minimizing the following loss

$$\mathcal{L}_\theta := \sum_{i=1}^{N} G(x_i, \psi_\theta(x_i), \nabla \psi_\theta(x_i), \nabla^2 \psi_\theta(x_i))^2 \tag{8}$$

where $G(x, \psi(x), \nabla \psi(x), \nabla^2 \psi(x)) = 0$ is the PDE whose solution $\psi(x)$ we are trying to approximate and $x_i$ are points taken from the discretization of our domain. In (8), the function $\psi_\theta(x) := A(x) + F(x, N_\theta(x))$ is a candidate approximation which by construction satisfies the boundary condition, where $N_\theta(x)$ is a feedforward neural network. In order to ensure that the conditions at the boundary are satisfied, $F(x, N_\theta(x)) = 0$ at the boundary and $A(x)$ is a fixed function which satisfies them.

Although this approach is well suited for some problems, special care must be taken when computing the gradient of the loss with respect to the parameters. For instance, following the previous procedure, the loss for HJI PDE would be written as

$$\mathcal{L}_\theta := \sum_{i=1}^{N} (\frac{\partial V(x_i, t_i)}{\partial t} + min\{0, H(x_i, \nabla_x V)\})^2, \tag{9}$$

but the $min$ in the function makes this expression not differentiable everywhere. There exist ways to circumvent this problem (Djeridane and Lygeros, 2006), however, but they require the cumbersome definition of many intermediary functions which can become hard to find for complicated dynamical models.

In this work, we try to tackle the problem of finding an approximate solution to (2) from a different perspective. We show that a poor approximation to our solution is enough to generate "good enough" new data for regression, which can in turn be used to improve our model.

## 4  SELF-GENERATED DATA

### 4.1  ALGORITHM

In this section we present a simple method for approximating the solution to (2) by utilizing a feedforward neural network in two ways: as a function approximator and a data generator. We believe that this parametric approach is better suited for finding good approximations by avoiding some of the limitations found in gridding/tabular techniques due to the curse of dimesionality. To

that end, we start by defining our candidate approximation $\hat{V}_\theta(x)$ to be of the same form as in (Lagaris et al., 1998); that is, a sum of two terms which help satisfy our boundary condition $V(x, 0)$

$$\hat{V}_\theta(x, t) = V(x, 0) + t\mathcal{N}_\theta(x, t), \tag{10}$$

where $\mathcal{N}_\theta(x, t)$ is a neural network mapping from our states and time variables to the real numbers. Next, we sample $N$ points in the state variable $x$ chosen uniformly at random over some set $S$ which includes $\mathcal{T}$ (the target set), and similarly, sample $N$ points in the time variable $t$ uniformly at random over the set $[-T, 0]$, where $T \geq 0$ is the desired time horizon. By sampling from these distributions, we seek to find a good approximation to $V(x, t)$ over the set $S \times [-T, 0]$. Once these points have been gathered, we make use of the update (4),(5) or (6) (depending on our problem) and use $\hat{V}_\theta(x, t)$, the approximation itself, to generate the new regression points. The complete algorithm 4.1 is shown using update equation (6), but it should be clear how to modify it for the other cases.

---

**Algorithm 1** Recursive Regression via SGD with Momentum

---

1: **Input:** $N$, interval, $V(x, 0)$, $f(x, a, b)$, $\mathcal{A}$, $\mathcal{B}$, $S$, T, $K$(batch size), $\gamma$ (momentum decay), $\eta$ (learning rate)
2: Random initialization of the weights and biases of the neural network $\mathcal{N}_\theta(x, t)$
3: Define $\hat{V}_\theta(x, t) := V(x, 0) + t\mathcal{N}_\theta(x, t)$
4: Define $\mathcal{L}_\theta := \sum_{k=0}^{K} |y_k - \hat{V}_\theta(x_k, t_k)|$
5: $i \leftarrow 0, \nu_i \leftarrow 0, \Delta t \leftarrow 10^{-2}$
6: **while** True **do** (or stopping criterion)
7: **if** mod(i,interval) == 0 **then**
8: $R \leftarrow$ empty array of size $N$
9: Sample $N$ pairs $(x, t) \sim Uniform(S \times [-T, 0])$
10: **for** $j = 0$ to $N$ **do**
11: $(a_j^*, b_j^*) \leftarrow \underset{a \in \mathcal{A}}{argmax} \, \underset{b \in \mathcal{B}}{argmin} \, \nabla_x \hat{V}_\theta^T f(x, a, b)$
12: $y_j \leftarrow min\{\hat{V}_\theta(x_j, t_j), \hat{V}_\theta(x_j + f(x_j, a^*, b^*)\Delta t, t_j) \}$
13: $R_j \leftarrow ((x_j, t_j), y_j)$
14: $b \leftarrow K$ elements from $R$ picked at random
15: $\nu_{i+1} \leftarrow \gamma \nu_i + \eta \nabla_\theta \mathcal{L}(b)$
16: $\theta_{i+1} \leftarrow \theta_i - \nu_{i+1}$
17: $i \leftarrow i + 1$
18: **Output:** $\hat{V}_\theta(x, t)$

---

## 4.2 Comments

Algorithm 4.1 is a type of bootstrapping method in that lines 12 and 13 make use of $\hat{V}_\theta(x, t)$ to generate points for regression to train $\mathcal{N}_\theta(x, t)$ which in turn modify $\hat{V}_\theta(x, t)$ itself. At first glance, it is unclear whether the generated pairs $((x_j, t_j), y_j)$ will result in a good approximation to the solution of our PDE after regression; however, given the form of our candidate function (10) we expect that points sampled near $t = 0$ will in fact be reasonable approximations of $V(x, t)$ for small $t$. Given this assumption, we hypothesize that despite the presence of misleading data, our network will be able to do a good job at regressing over all points, thus improving our initial model and allowing the generation of improved data. By repeating this procedure, we expect the accuracy of the boundary condition to "propagate" backward in time (possibly with some minor error) in the form of better and better points for regression.

Another important aspect from line 13 is that we are simulating our dynamics forward in time using the Euler approximation step $x_j + f(x_j, a^*, b^*)\Delta t$. In practice, depending on the variability and complexity of the dynamics, one might use a Runge-Kutta method or a more involved integration procedure. For the experiments in the next sections a Runge-Kutta method with 4 stages (RK4) was used.

## 5 EXPERIMENTS

In this section we present a few 2-dimensional experiments to demonstrate the validity of our claim and the effectiveness of the algorithm. To measure the performance of the algorithm, we compare the difference between our computed approximation and the true analytical solution. In case it is not straightforward to obtain the solution, a very accurate approximation taken from state-of-the-art tools is used instead. In particular, we make use of the LevelSet Toolbox from Mitchell (2007), a powerful computational tool for obtaining good approximations to Hamilton-Jacobi (HJ) PDEs.

The first error metric to be used will be

$$E_1(\hat{V}_\theta(x,t)) := \frac{1}{M} \sum_{i=1}^{M} |V(x_i,t_i) - \hat{V}_\theta(x_i,t_i)| \tag{11}$$

where $M$ are the number of points chosen from our domain to compute the average absolute error and $V(x,t)$ can denote either the true solution or an accurate approximation. In the case where the analytical solution is known, the points are taken uniformly at random over $S$; otherwise, they are taken over some grid in $S$ and $[-T, 0]$. Lastly, we also use a second error metric

$$E_2(\hat{V}_\theta(x,t)) := \frac{1}{M} \sum_{i=1}^{M} |\frac{\partial V(x_i,t_i)}{\partial t} + min\{0, H(x_i, \nabla_x V)\}| \tag{12}$$

similar to the one defined in (9), which denotes the extent by which (on average) the approximation is violating the PDE equality. For all experiments $M = 3000$, all chosen uniformly at random over $S \times [-T, 0]$. In section 5.4 we also show a visual representation of the approximations.

### 5.1 A LINEAR SYSTEM

In this experiment we study the performance of the algorithm on an autonomous system of the form

$$\dot{x} = f(x) = \begin{bmatrix} -1 & -2 \\ 2 & -1 \end{bmatrix} x \tag{13}$$

with $V(x,0) = ||x||_2 - 1$ and $T = 1.0$. For this simple system, the solution to the HJI PDE can be found analytically to be $V(x,t) = e^{-t}||x||_2 - 1$. One can easily verify this by checking it satisfies the boundary condition and (2). For this experiment, a feedforward neural network with a single hidden layer of 10 units and sigmoid activation functions was used. The number of points sampled was chosen to be $N = 500$, uniformly picked over the set $S := \{(x_1, x_2)|x_1, x_2 \in [-5, 5]\}$ and over $t \in [-T, 0]$. The batches were picked to be of size $K = 10$, momentum decay $\gamma = 0.95$ and learning rate $\eta = 0.1$. The interval to renew the regression points was chosen to be 1000 iterations and the program was halted at 500,000 iterations.

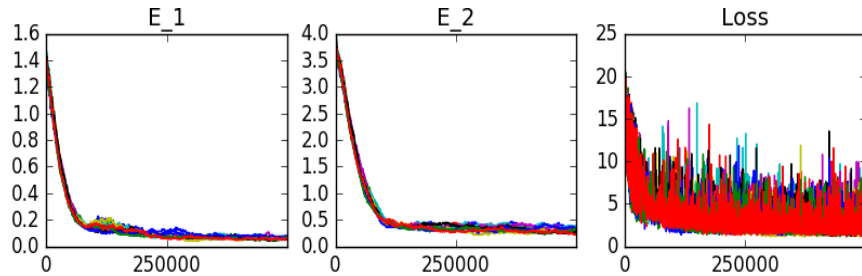

Figure 1: From left to right: the first figure shows the mean absolute error $E_1$, the second figure shows the mean absolute PDE error $E_2$ and the third figure shows the loss $\mathcal{L}_\theta$ as defined in algorithm 4.1 over all the data. The horizontal axis represents the iteration number.

The results shown in Fig. 1 where taken over 10 runs of the algorithm concurrently executed over multiple threads. The overall time to run the 500,000 iterations for all threads was 1521 seconds. The average $E_1$ error at halting time was in the order of $7 \times 10^{-2}$, whereas the $E_2$ error was in the order of $3 \times 10^{-1}$. The sharp jumps appearing in the loss figure in the majority of cases correspond to the error after new points are generated and used for regression.

## 5.2 PURSUIT-EVASION GAME: SINGLE INPUT

In this experiment we explore a pursuit-evasion game where a pursuer has to intercept an evader. In a first simplified approach, we assume the evader has a fixed heading and speed, whereas the pursuer has the same speed as the evader but has the liberty to change the direction of its heading. Fixing the evader at the origin with its heading aligned with the x-axis we frame the problem in relative coordinates between the evader and pursuer, that is $x = [x_r \ y_r]^T$, where $x_r$ and $y_r$ represent the $x$ and $y$ position of the pursuer relative to the evader. This system's dynamics are readily encoded in the following equation

$$\begin{bmatrix} \dot{x}_r \\ \dot{y}_r \end{bmatrix} = f(x, b) = \begin{bmatrix} v_p cos(b) - v_e \\ v_p sin(b) \end{bmatrix} \tag{14}$$

where $v_p = v_e = 2.0$ represent the speed of the pursuer and evader respectively, $b \in [0, 2\pi]$ represents the input available to the pursuer, which is the angle with respect to the x-axis. In this simplified pursuit-evasion game we say the pursuer has captured the evader if they are within 1 unit of distance from each other. Thus, we define our capture condition by defining $V(x, 0) = ||x||_2 - 1$, which will ensure that our approximation captures all the states from which the pursuer can capture the evader in within $T = 1.0$. As in the previous example, we choose the same network architecture and the same values for the halting time, renewal interval, $N, K, \gamma$ and $\eta$.

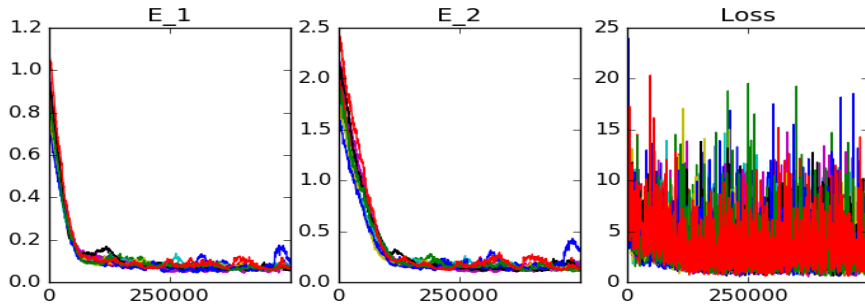

Figure 2: From left to right: the first figure shows the mean absolute error $E_1$, the second figure shows the mean absolute PDE error $E_2$ and the third figure shows the loss $\mathcal{L}_\theta$ as defined in algorithm 4.1 over all the data. The horizontal axis denotes iteration number.

The results shown in Fig. 2 where also taken over 10 runs of the algorithm like in section 5.2. The overall time to run the 500,000 iterations was 1952 seconds. The average $E_1$ error at halting time was also in the order of $7 \times 10^{-2}$, whereas the $E_2$ error was in the order of $1.5 \times 10^{-1}$. The points used to compute $E_1$ were taken from a $51 \times 51$ grid at $t = -0.5$ (half of the time horizon), using a previously computed approximation from the LevelSet Toolbox. The reason why a single time instance was used to compute $E_1$ was purely to reduce the amount of computation of the error at run-time.

## 5.3 PURSUIT-EVASION GAME: TWO INPUTS

The last experimental example also consists of a pursuit-evasion game, but in this case the evader has access to a range of speeds through an input $a \in [-2, 2]$. The system dynamics thus become

$$\begin{bmatrix} \dot{x}_r \\ \dot{y}_r \end{bmatrix} = f(x, a, b) = \begin{bmatrix} v_p cos(b) - a \\ v_p sin(b) \end{bmatrix} \tag{15}$$

and, similarly, $V(x,0) = ||x||_2 - 1$ and $T = 1.0$. As before, $v_p = 2.0$. The interesting behavior we expect to see from this experiment, in comparison to the single input counterpart, is that this new available action to the evader will make it more difficult for the pursuer to intercept. This should then be evident by looking at our approximation $\hat{V}_\theta$ and its zero sub-level sets at different times. For this experiment we also chose the same architecture for the network as in the previous experiments and the same parameters, except for the halting time which was 300,000 iterations.

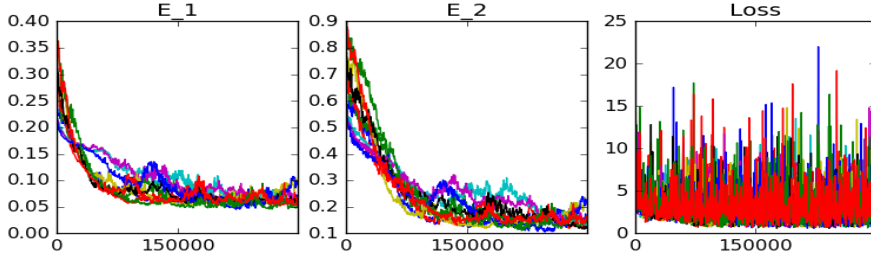

Figure 3: From left to right: the first figure shows the mean absolute error $E_1$, the second figure shows the mean absolute PDE error $E_2$ and the third figure shows the loss $\mathcal{L}_\theta$ as defined in algorithm 4.1 over all the data.

The results shown in Fig. 3 where also taken over 10 runs of the algorithm. The overall time to run the 300,000 iterations over the all threads was 1028 seconds. The average $E_1$ error at halting time was in the order of $6 \times 10^{-2}$, whereas the $E_2$ error was in the order of $1.5 \times 10^{-1}$. Like in the single input case, the points used to compute $E_1$ were taken from a $51 \times 51$ grid at $t = -0.5$ of a pre-computed approximation.

## 5.4 CONTOUR VISUALIZATION

In this section we briefly display some of the contours for a neural network picked at random from those computed in the experimental section. Each line corresponds to the set of states where $\hat{V}_\theta(x,t) = 0$ for $t = 0, -0.25, -0.5, -0.75, -1.0$. These contours enclose within them the states from which our system can reach the target set $\mathcal{T}$ within the absolute value of its associated time.

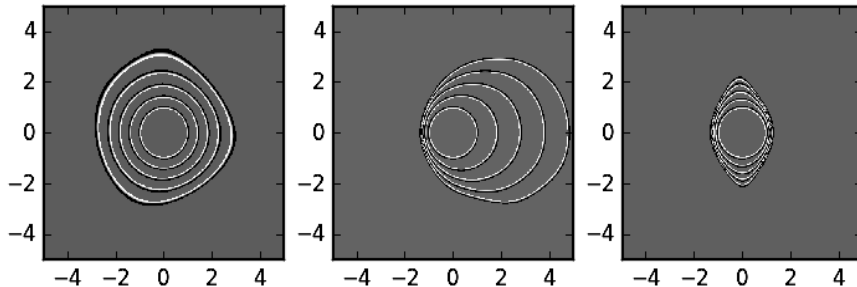

Figure 4: From left to right: contours for experiment one, experiment two and experiment three. As one can appreciate, the contours grow according to the specified dynamical model.

As expected, the linear system's contours expand radially in all directions since the origin is a stable equilibrium point[3] where all trajectories converge. For the pursuit-evasion game of one input, we also see that the contours grow toward the right, which is a sensible outcome given that the pursuer can't catch up with the evader if it starts somewhere where $x_r < -1.0$. Finally, the last set of contours associated with the pursuer-evader game of two competing inputs also make sense, since starting states $x_r < -1.0$ or $x_r > 1.0$ should not permit the pursuer to intercept the evader, and so

---

[3] with the same negative real part for the eigenvalues

the contours should not expand in those directions. As a last comparison, in Fig. 5 we display the actual contours that would be obtained using the LevelSet Toolbox.

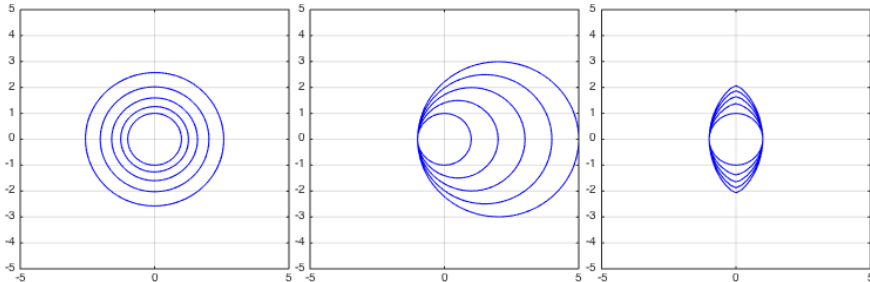

Figure 5: Contours obtained from the LevelSet Toolbox in Matlab.

By comparing Fig. 5 and 4 one can qualitatively see that the neural network has learned an accurate approximation of $V(x,t)$.

## 6 ADVANTAGES AND DISADVANTAGES

The first advantage of using this method over gridding techniques is a dramatic improvement in memory requirements. For instance, using a standard grid with $[51, 51, 10]$ discretization points per axis (i.e. 51 in $x_r$, 51 in $y_r$ and 10 in t) each of the three previous experiments requires the storage of $26,010$ numbers, as opposed to 51 weights for our neural network. For the gridding approach this memory requirement must increase exponentially with the number of dimensions, whereas this need not be the case for our method. Furthermore, points that do not fall exactly on the grid have to be interpolated, whereas the neural network is an approximation that assigns values to all points in the domain. To this we can also add that fact that the neural network can yield the gradient at any point directly with backpropagation, whereas the gradient must once again be approximated for gridding techniques.

The main disadvantage of this method, for small dimensional systems in particular, is the time requirement. Computing values over a grid with the LevelSet Toolbox for the previous systems took less than 10 seconds. This advantage of gridding/tabular procedures, however, quickly disappears in higher dimensions (4D, 5D...) due to the curse of dimensionality. Finally, another disadvantage of using this method is the necessity to tune hyper parameters.

## 7 CONCLUSION AND FUTURE WORK

In this work we focus our attention on the idea that recursive/bootstrapped regression can be used in some problems where the function we wish to approximate has some known characteristics. In particular, we show that accurate approximations to the HJI PDE solution can be found by assigning a neural network two roles, one of them being function approximation, and the other data generation.To validate our hypothesis three different experiments with three distinct dynamical systems were performed with satisfactory results.

In this work we did not focus on the architecture of the neural network, but rather on its ability to perform well on three distinct tasks using the same algorithm. In future work we will try to find whether one can construct wider or deeper neural networks and obtain better results. We also want to investigate how well this method scales with the number of state and input dimensions. Positive results in that front could suppose an important step to further alleviate the effects of the curse of dimensionality, which are pervasive in griding methods.

## ACKNOWLEDGMENTS

Special thanks to Carlos Florensa for his implementation tips and to Jaime F. Fisac for helping in the process of writing this work.

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

## 8  EXTRA EXPERIMENT

This experiment was designed to test the applicability of the method to problems beyond those presented in the previous sections. In particular, we show that with small changes we can also compute an accurate approximation to a pursuit-evasion problem in 3 dimensions. Similar to the previous examples, we frame the problem in relative coordinates with the x-axis aligned with the evader's heading, and give the pursuer and evader control over the rate of rotation. This can be written as follows:

$$\begin{bmatrix} \dot{x}_r \\ \dot{y}_r \\ \dot{\theta}_r \end{bmatrix} = f(x,a,b) = \begin{bmatrix} -v_e + v_p cos(\theta_r) + ay_r \\ v_p sin(\theta_r) - ax_r \\ b - a \end{bmatrix} \tag{16}$$

For this problem the capture condition is encoded in the boundary condition $V(x,0) = ||[x_r \ y_r]^T||_2 - 1$ (where we ignore $\theta_r$ since the capture condition only depends on the distance) and we consider a the time horizon $T = 1.0s$. For this problem we give both pursuer and evader the same speed $v_p = v_e = 1.0$ and the same turning rates $a, b \in [-1, 1]$. Unlike the previous experiments, we used a neural network with two hidden layers with 10 and 5 units respectively and sigmoid activations. The number of points sampled was chosen to be $N = 2000$, uniformly picked over the set $S := \{(x_r, y_r, \theta_r) | x_r, y_r \in [-5, 5], \theta_r \in [-\pi, \pi]\}$ and over $t \in [-T, 0]$. The batches were picked to be of size $K = 25$, momentum decay $\gamma = 0.999$ and learning rate $\eta = 0.001$. The interval to renew the regression points was chosen to be 1000 iterations and the program was halted at 500,000 iterations.

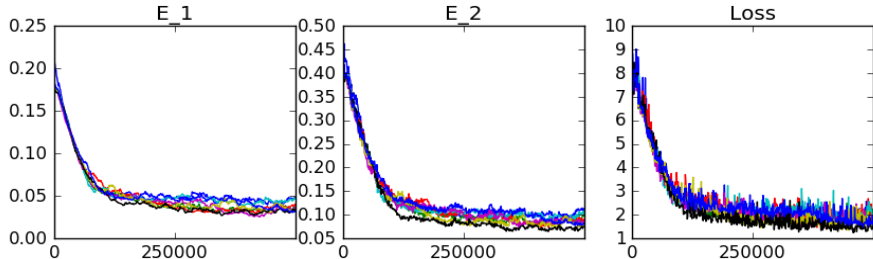

Figure 6: From left to right: the first figure shows the mean absolute error $E_1$, the second figure shows the mean absolute PDE error $E_2$ and the third figure shows the loss $\mathcal{L}_\theta$ as defined in algorithm 4.1 over all the data.

As shown in Fig. 6, both error metrics decrease as the algorithm progresses, reaching an average error for $E_1$ in the order of $5.0 \times 10^{-2}$ and an average error for $E_2$ in the order of $1.0 \times 10^{-1}$. The points used to compute $E_1$ were taken from a $51 \times 51 \times 50$ approximation grid at $t = -0.5s$. This set of experiments was run in a different machine[4] using 8 threads and the total time for all threads to finish was 1000 seconds. Finally, Fig. 7 shows the zero level set contour at $t = -0.5$, which is now a 3D surface, from side and top perspectives. The first row shows the output of the LevelSet Toolbox from each perspective, and the second row shows a 3D scatter plot of points on the zero level-set obtained from one of the 8 neural networks that were trained.

---

[4]due to heavy usage of the first machine we had to switch to a different one

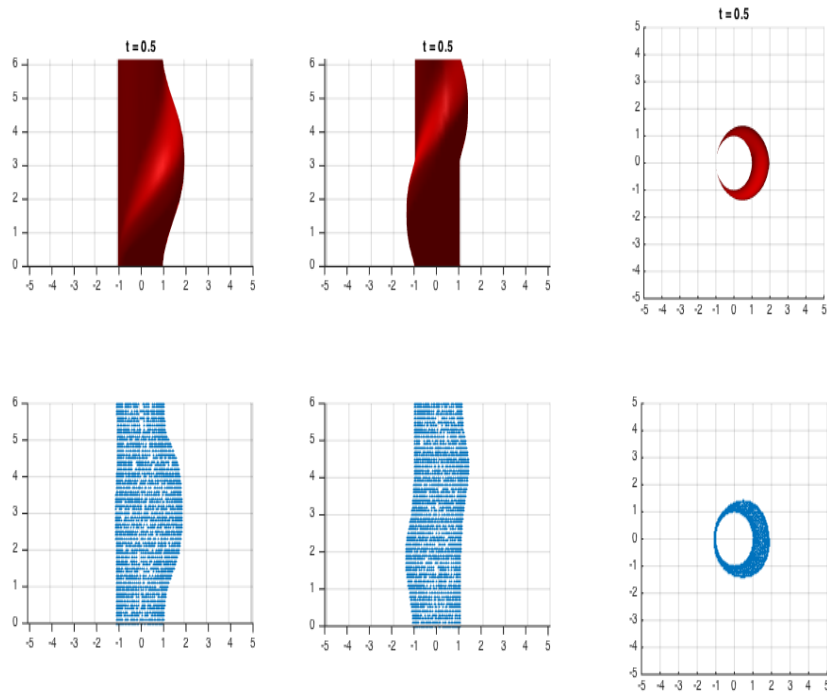

Figure 7: The first column shows the first side view perpendicular with respect to the x-z plane. The second column shows the second side view perpendicular with respect to the y-z plane. Finally, the third column shows the top view which is perpendicular with respect to the x-y plane.

For this experiment, only 111 numbers were needed to store the approximation, as opposed to $51 \times 51 \times 50 \times 10 = 1300500$ numbers (i.e. 51 in $x_r$, 51 in $y_r$, 50 in $\theta_r$ and 10 in t) for a $[51 \times 51 \times 50 \times 10]$ grid approximation.

