# Peer review of "Recursive Regression with Neural Networks: Approximating the HJI PDE Solution"

_ICLR 2017 — rejected_

[Official Review · AnonReviewer1 · rating 5 · confidence 1 · 14 Dec 2016]
**Hard to follow; unclear about contribution**

I have no familiarity with the HJI PDE (I've only dealt with parabolic PDE's such as diffusion in the past). So the details of transforming this problem into a supervised loss escape me. Therefore, as indicated below, my review should be taken as an "educated guess". I imagine that many readers of ICLR will face a similar problem as me, and so, if this paper is accepted, at the least the authors should prepare an appendix that provides an introduction to the HJI PDE. At a high level, my comments are:

1. It seems that another disadvantage of this approach is that a new network must be trained for each new domain (including domain size), system function f(x) or boundary condition. If that is correct, I wonder if it's worth the trouble when existing tools already solve these PDE's. Can the authors shed light on a more "unifying approach" that would require minimal changes to generalize across PDE's?

2. How sensitive is the network's result to domains of different sizes? It seems only a single size 51 x 51 was tested. Do errors increase with domain size?

3. How general is this approach to PDE's of other types e.g. diffusion?

[Official Review · AnonReviewer3 · rating 3 · confidence 5 · 18 Dec 2016]
**somewhat interesting paper, wrong conference**

Approximating solutions to PDEs with NN approximators is very hard. In particular the HJB and HJI eqs have in general discontinuous and non-differentiable solutions making them particularly tricky (unless the underlying process is a diffusion in which case the Ito term makes everything smooth, but this paper doesn't do that). What's worse, there is no direct correlation between a small PDE residual and a well performing-policy [tsitsiklis? beard? todorov?, I forget]. There's been lots of work on this which is not properly cited. 

The 2D toy examples are inadequate. What reason is there to think this will scale to do anything useful? 

There are a bunch of typos ("Range-Kutta"?) .

More than anything, this paper is submitted to the wrong venue. There are no learned representations here. You're just using a NN. That's not what ICLR is about. Resubmit to ACC, ADPRL or CDC.

Sorry for terseness. Despite rough review, I absolutely love this direction of research. More than anything, you have to solve harder control problems for people to take notice...

[Official Review · AnonReviewer2 · rating 7 · confidence 3 · 20 Dec 2016 (modified: 26 Jan 2017)]
**Good Work, Preliminary Results**

This paper presents an algorithm for approximating the solution of certain time-evolution PDEs. The paper presents an interesting learning-based approach to solve such PDEs. The idea is to alternate between:
1. sampling points in space-time
2. generating solution to PDE at "those" sampled points
3. regressing a space-time function to satisfy the latter solutions at the sampled points (and hopefully generalize beyond those points).

I actually find the proposed algorithm interesting, and potentially useful in practice. The classic grid-based simulation of PDEs is often too expensive to be practical, due to the curse of dimensionality. Hence, learning the solution of PDEs makes a lot of sense for practical settings. On the other hand, as the authors point out, simply running gradient descent on the regression loss function does not work, because of the non-differentiablity of the "min" that shows up in the studied PDEs.

Therefore, I think the proposed idea is actually very interesting approach to learning the PDE solution in presence of non-differentability, which is indeed a "challenging" setup for numerically solving PDEs.

The paper motivates the problem (time-evolution PDE with "min" operator applied to the spatial derivatives) by applications in control thery, but I think there is more direct interest in such problems for the machine learning community, and even deep learning community. For example

[Author Response · Vicenç Rubies Royo · 13 Jan 2017]
**Extra Experiment Appended**

We appended a 3D experiment in the paper.

[Final Decision · Program Chairs · 06 Feb 2017]
**ICLR committee final decision**

The basic approach of this paper is to use a neural net to sequentially generate points that can be used as the basis points in a PDE solver. The idea is definitely an interesting one, and all three reviewers are in agreement that the approach does seem to have a lot of potential.
 
 The main drawback of the paper, simply, is that it's unclear whether this result would be of sufficient interest for the ICLR audience. Ultimately, it seems as though the authors are simply training a neural network to generate this points, and the interesting contribution here comes from the application to PDE solving, not really from any advance in the NN/ML side itself. As such, it seems like the paper would be better appreciated (as a full conference paper or journal paper), within the control community, rather than ICLR. However, I do think that as an application, many at ICLR would be interested in seeing this work, even if its likely to have relatively low impact on the community. Thus, I think the best avenue for this paper is probably as a workshop post at ICLR, hopefully with further submission and exposure in the controls community.
 
 Pros:
 + Nice application of ML to a fun problem, generating sample points for PDE solutions
 + Overall well-written and clearly presented
 
 Cons:
 - Unclear contribution to the actual ML side of things
 - Probably better suited to controls conferences